# Spatial Analysis of Mangrove Forest Management to Reduce Air Temperature and CO₂ Emissions

**Sumarmi Sumarmi** *, **Purwanto Purwanto and Syamsul Bachri**

Department of Geography, State University of Malang, Malang 65145, Indonesia; purwanto.fis@um.ac.id (P.P.); syamsul.bachri.fis@um.ac.id (S.B.)
*   Correspondence: sumarmi.fis@um.ac.id

**Abstract:** Surabaya is a coastal city that is still developing. Like other developing cities, Surabaya highly suppresses mangrove forests for residential, industrial, and other areas. Mangrove forests supply oxygen for the population of Surabaya. Forest mangroves reduce the effects of global warming and preserve sustainable coastal ecosystems. This research aimed to (1) map temperature changes in Surabaya over a period of 20 years (1996–2016) by using remote sensing and GIS, and (2) examine mangrove forests' ability to absorb $CO_2$ and decrease the impact of global warming in Surabaya. Research results showed that: (1) on the basis of the analysis of the temperature surface area, temperatures changed significantly between 1996 and 2016. Temperature changes can be classified into low, moderate, or high. The low-temperature area of 21–30 °C followed a different pattern. Each year, changes in the high-surface-temperature area were in the range of 31–42 °C. Changes highly increased in the period of 2006–2016. This indicates that Surabaya experienced a significant temperature increase in 2016. (2) There was correlation between the change in mangrove forest cover and the change in temperature; $CO_2$ concentration in mangrove, vegetation, and water areas decreased as it grew in areas used for construction, such as factories, residences, and roads. $CO_2$ concentration in Surabaya showed a distribution in the "high" and "extremely high" categories. The high category was 27.5%, and the extremely high category was 67.5%. The sample point in both the moderate and low category was around 25%.

**Keywords:** spatial analysis; mangrove forest; $CO_2$ emissions

## 1. Introduction

Certain cities in Indonesia have a higher average surface temperature than that of the surrounding areas due to the high level of urban physical development, including commercial, governmental, residential, and industrial buildings. A vegetation zone map revealed a decrease in vegetation zones such as paddy fields, plantations, and forest protection areas [1].

Increasing temperatures were documented from 1994 to 2012 in the city of Surabaya, Indonesia. According to Bryan [1], cumulative average temperatures from 1994 to 2002 were ranged from 28.95 to 38.23 °C. The recorded temperature of 2006 and 2009 also increased from 43.06 to 44.30 °C. However, the temperature recorded in 2012 declined to 34.81 °C. Once the $CO_2$ concentration in the atmosphere hit 450 ppm, the temperature rose by more than 2 °C [2].

The construction of Juanda Airport and residential and industrial areas in the eastern part of Surabaya has placed mangrove forest preservation at risk. Mangrove forests produce oxygen for Surabaya and absorb solar radiation to reduce the impact of global warming.

Mangrove forests contain a higher level of carbon than that of other kinds of forests on Earth [3]. The mangrove ecosystem in Indonesia is able to absorb carbon content in the air of up to 67.7 MtCO per year. Rachmawati [4] stated that mangroves have a biomass potential of 108.66 ton/ha and carbon dioxide concentration of 55.35 ton/ha. The high

levels of carbon dioxide are influenced by mangroves' ability to absorb carbon from their surroundings through a photosynthetic process known as sequestration [5]. During the photosynthetic process, carbon dioxide released by the atmosphere is bound by the plants and stored as biomass, resulting in a decrease in the carbon dioxide concentration in the air [6].

Mangrove forests have a higher capacity for carbon assimilation, absorption, and storage than that of other tropical forests [3]. The carbon content of mangroves is the highest in tropical areas with 1023 Mg C/Ha (above-ground carbon reserve). The absorbed carbon biomass varies with the age, species, morphology, and location of mangroves [7].

Mangrove forests play a role in reducing the heat caused by climate change, especially through carbon sequestration [8]. Carbon sequestration by mangroves is unique, and mangrove forests play an essential role in absorbing atmospheric carbon to control environmental ecosystems in coastal areas [9].

Plants can radiate and absorb electromagnetic waves. The ability of mangrove forests to absorb and reflect the electromagnetic spectrum makes it easy to tell mangrove vegetation apart from non-mangrove vegetation. Mangrove forests are highly effective in absorbing the electromagnetic spectrum between 400 and 1000 nm [10,11], with high sensitivity. The mangrove forest vegetation index is recognized at *NDVI* values ranging from –1 to 1, but its values generally range from 0.1 to 0.7 [12]. This is supported by a study that analyzed the reflectance of three distinct types of mangrove leaves during rainy and dry seasons [10]. Furthermore, the reflectance of mangrove species significantly varies.

Therefore, this can be used to refer to waves radiated by other objects, allowing for vegetation to be distinguished from other objects [13]. The analysis can be manually performed or carried out with the use of a computer. The LanduseSim technique can be utilized for spatial simulation, including land-use simulation to accommodate bottom-up and top-down approaches [14].

Besides forecasting, remote-sensing technology can present data and facts of past spatial issues. Spatial information is gathered with spatiotemporal analysis, which shows multilevel, multispectral, multitemporal images with a larger spatial extent.

Remote sensing is a scientific method that involves obtaining information about an object, location, or symptom by evaluating the sensor's data without requiring physical contact [15]. Remote sensing is used in various disciplines, including cartography, agriculture, forestry, natural-resource management, urban and regional planning, and other Earth sciences [16,17].

One of the applications of remote sensing is the ability to examine and evaluate changes in mangrove forests on a spatial level. Therefore, spatial analysis of changes in mangrove areas is essential to determine the $CO_2$ concentration in mangrove forests.

## 2. Materials and Methods

This research aimed to (1) map temperature changes in the Surabaya Regency over a period of twenty years (1996–2016) by using remote sensing and GIS, and (2) examine mangrove forests' ability to absorb $CO_2$ and decrease the impact of global warming in Surabaya.

This research consisted of a survey designed using remote sensing and a geographic information system (GIS). Remote-sensing data from Landsat and Quickbird satellite images were used to identify temporal changes in mangrove forests and temperature over a period of 20 years. The data were then combined with field data and analyzed using GIS to determine the mangrove-area distribution and temperature patterns in Surabaya. Remote-sensing data obtained from Quickbird were used to examine the existing land use with the activities of the local community in mangrove areas.

Landsat is an automated Earth resource satellite equipped with a nonphotographic sensor. The most recently launched Landsat satellite, on 15 April 1999, was Landsat 7, which carries an Enhanced Thematic Mapper Plus (ETM+) sensor. The characteristics of Landsat ETM+ are shown in Table 1 below.

**Table 1.** Characteristics of Landsat-7 ETM+.

| System | Landsat 7 |
|---|---|
| Orbit | 705 km, 98.2°, sun-synchronous, passes over at 10.00 a.m. once every 16 days |
| Sensor | Enhanced Thematic Mapper (ETM+) |
| Coverage | 185 km (FOV = 15°) |
| Side view | Not available |
| Temporal resolution | 16 days |
| Used wavelength channels (μm) | (Band 1) 0.45–0.52 μm, (Band 2) 0.52–0.60 μm, (Band 3) 0.63–0.69 μm, (Band 4) 0.76–0.90 μm, (Band 5) 1.55–1.75 μm, (Band 6) 10.4–12.50 μm, (Band 7) 2.08–2.34 μm, (PAN) 0.5–0.9 μm |
| Spatial resolution | (PAN) 15 m, (bands 1–5, 7) 30 m, and (Band 6) 60 m |
| Data acquisition | https://earthexlorer.usgv.gov/, accessed on 21 September 2001 |

Source: Landsat Project Science Office, 2002.

The purpose of this research was to examine changes in the mangrove area of East Surabaya due to global warming. The tools consisted of a computer, an ENVI/ErMapper to analyze satellite images, SIG software equipped with ArcGIS 10.2 for global positioning system (GPS) spatial-data analysis, LanduseSim 2.2 software, a thermometer to check the field temperature, and a camera. Materials included topographic maps of Indonesia at scales of 1:25,000 and 1:10,000, a bathymetric/coastal environment map of Indonesia at a scale of 1:50,000, and spatial-planning maps of Surabaya and Sidoarjo. Landsat MSS, TM, ETM+, Landsat 8, DEM Aster, and Quickbird satellite images were used.

The collected data were categorized into primary and secondary data. Primary data were data of field temperature, existing land use, and the examination of mangrove forests based on the ability to absorb $CO_2$ and heat energy. Secondary data contained satellite images and maps collected through survey and documentation. The survey was designed to directly observe the research subjects, namely, temperature, land use, and the amount of absorbed $CO_2$ and heat energy. Documentation was performed to collect the data according to the actual spatial use as authentic evidence.

Analysis of changes in mangrove forests was performed via the normalized difference vegetation index (*NDVI*) method, which is a standard method for comparing the level of vegetation of satellite data. The formula for the *NDVI* is as follows:

$$NDVI = (NIR - Red)/(NIR + Red). \tag{1}$$

The mangroves' ability to absorb heat energy could be identified from temperature changes, especially in the mangrove area located in East Surabaya for a period of 20 years. Thermal infrared channels in Channel 6 of Landsat TM and ETM+ and Channels 10 and 11 in Landsat 8 were used to observe the temperature.

Analyses used to study the effect of changes in land cover on temperature were Landsat 7 image analysis and simple linear regression. The results obtained from observation and field measurement were analyzed to then be explained or described. Landsat 7 image analysis was conducted by converting imagery data into temperature data. The converted imagery data were pixel values in Band 6 of the Landsat images, called digital numbers (DNs). There are two necessary steps to convert imagery data into temperature data [15]:

1.  Conversion of a digital number (*DN*) into spectral radiance ($L_\lambda$): Formula:

$$Radiance\ (L_\lambda) = (gain \times DN) + offset \tag{2}$$

where:

$L_\lambda$ = spectral radiance in watts;
*gain* = a constant of 0.05518;
*DN* (digital number) originating from pixel values of the images; and

*offset* is a constant of 1.2378.

2.  Spectral-radiance conversion into temperature. The spectral radiance on a thermal band image (band 6) could then be converted into temperature. The equation of spectral-radiance conversion into temperature is:

$$T = \frac{K2}{ln\left(\frac{K1}{L_\lambda + 1}\right)} \tag{3}$$

where:

$T$ = temperature;
$K1$ = constant in watts with a value of 666.09 ETM+ and 607.76 for TM;
$K2$ = constant in Kelvin with a value of 1282.71 for ETM+ and 1260.56 for TM; and
$L$ = spectral radiance in watts.

The final result obtained from the equation was in temperature on a Kelvin ($K$) scale. In order to obtain temperature data on a Celsius (°C) scale, the equation was changed into:

$$T = \left(\frac{K2}{ln\left(\frac{K1}{L_\lambda + 1}\right)}\right) - 273 \tag{4}$$

where:

$T$ = temperature in Celsius (°C).

Simple linear-regression analysis was performed when the types of the variables were free and bound. The land-cover area is an independent variable (X), while air temperature is a dependent variable (Y). Regression analysis was performed by using SPSS 16 software for Windows.

3.  The calculation of the biomass and carbon of the mangrove ecosystem was performed with the following formula [5]: Carbon and biomass potentials:

$$ni = 1 = \left(\frac{yi}{plot\ area}\right) \times \text{mangrove area} \tag{5}$$

where:

$yi$ = biomass potential per type; and
$i = i$ type.

4.  Spatial modeling with LanduseSim. Spatial modeling uses LanduseSim to predict changes in land use and temperature, and to determine the scenario of mangrove forest planning.

## 3. Results

### 3.1. Temperature Changes in Surabaya

Imagery data used in this research were Landsat 5 TM, Landsat 7 ETM+, and Landsat 8 OLI TIRS satellite images in *.GeoTIFF* format, which contained metadata in *.MTL* format. The metadata provided information on image recording time, solar elevation angle, image quality, cloud-coverage area, and other important data relevant to the images. The imagery data utilized in this research are shown in Table 2.

Several imagery data that were used in this research were Landsat Level 1 Standard Data Products. Level 1 Landsat data were imagery data resulting from the calibration and validation processes used to measure and prevent sensory problems such as the atmospheric condition so that their quality can be improved by using certain algorithms [4,18].

The study of surface temperature in Surabaya in a period of two decades was analyzed through Landsat TM, ETM+, and Landsat 8 images satellite of Operational Land Imager (OLI). The results of analysis are shown in Table 3 below.

**Table 2.** Research imagery data.

| No. | Name of Satellite Imageries | Data and Time of Acquisition | WRS PATH/ROW | Image Quality and Cloud Coverage |
|---|---|---|---|---|
| **1.** | LT51180651996211DKI00 | 29 July 1996/01:50:53 | 118/65 | 9/3.00 |
| **2.** | LE71180652001264DKI00 | 21 September 2001/02:24:13 | 118/65 | 9/10.00 |
| **3.** | LE71180652006246EDC00 | 3 September 2006/02:25:23 | 118/65 | 9/0.00 |
| **4.** | LE71180652011244EDC00 | 1 September 2011/02:29:07 | 118/65 | 9/2.00 |
| **5.** | LC81180652016090LGN00 | 30 July 2016/02:35:23 | 118/65 | 9/25.36 |

**Table 3.** Surface-temperature changes in Surabaya.

| Temperature Classes | Annual Temperature Change Area (km$^2$) | | | | |
|---|---|---|---|---|---|
| | **1996** | **2001** | **2006** | **2011** | **2016** |
| Low (11–21 °C) | 0.0657 | 0.4734 | 0.1071 | 5976 | 9296 |
| Moderate (22–31 °C) | 334,2231 | 111,4416 | 103,8618 | 323,2089 | 62,0028 |
| High (32–41 °C) | 0.0054 | 222,3792 | 230,319 | 5103 | 271,3618 |
| Total area | 334,2942 | 334,2942 | 334,2879 | 334,2879 | 334,2942 |

Temperature-change trends from 1996 to 2016 can be determined using analytical results of the surface temperature area. Trends could be classified into the three temperature characteristics of low, moderate, and high. From 1996 to 2016, the low temperature ranged between 2 and 11 °C. This indicates that Surabaya experienced a temperature increase in the low-temperature category. Temperature changes in low-temperature areas are shown in Figure 1.

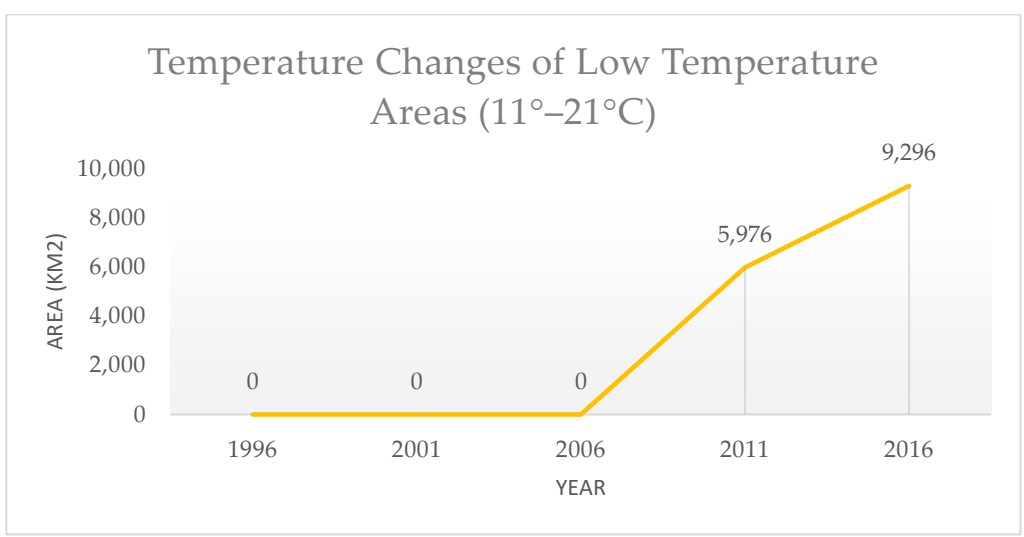

**Figure 1.** Temperature changes in low-temperature areas in Surabaya in 1996–2016.

Moderate temperature changes in the 21–30 °C range showed negative trends, which meant that areas with a moderate surface temperature declined annually. The temperature changes in moderate-temperature areas are shown in Figure 2.

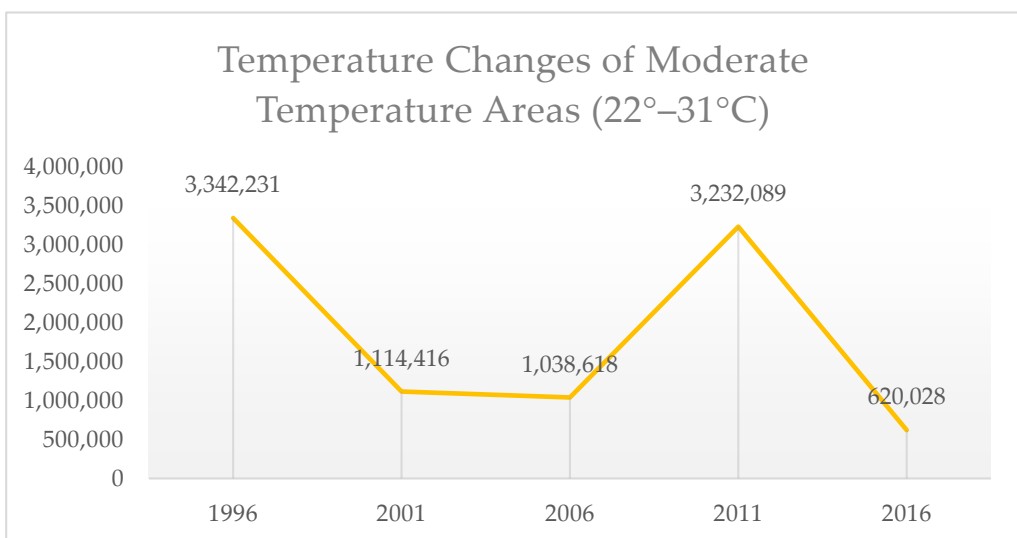

**Figure 2.** Temperature changes in moderate-temperature areas in Surabaya in 1996–2016.

The temperature fluctuated moderately between 21 and 30 °C. In early 1996, moderate surface temperatures dominated Surabaya and covered an area of 334.2231 km$^2$, but this decreased from 2001 to 2006. There was an area expansion from 2006 until 2011. However, a drastic decrease in 2016 resulted in a final area of 62.0028 km$^2$.

The high surface temperature of 31–42° C fluctuated. The fluctuating point of area expansion with high surface temperatures occurred between 1996 and 2001, and then decreased between 2001 and 2006 before increasing again from 2006 to 2016. This showed that high-temperature areas in Surabaya kept increasing in 2016. Temperature changes in high-temperature areas are shown in Figure 3.

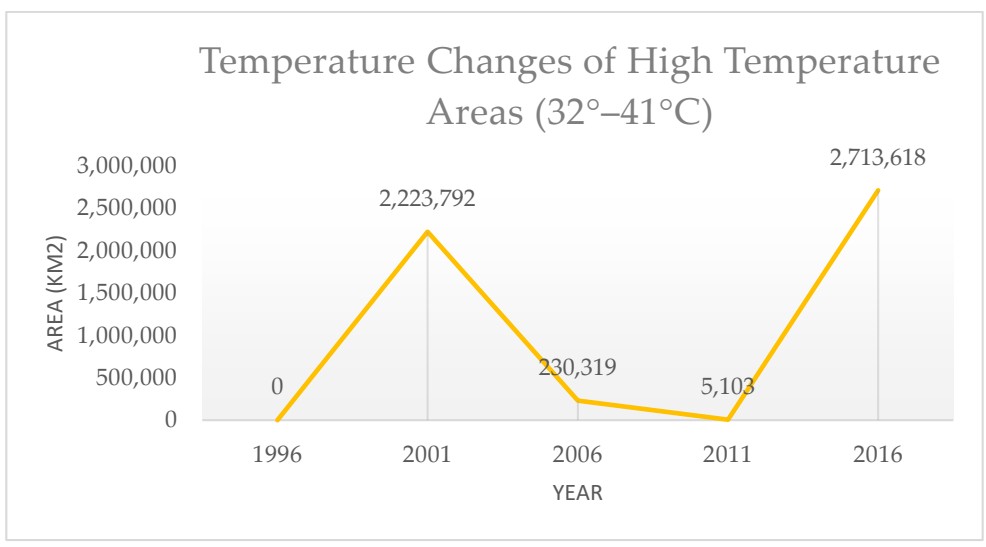

**Figure 3.** Temperature changes in high-temperature areas in Surabaya in 1996–2016.

Surface-temperature changes in Surabaya occurred in 1996–2016 due to various factors. Some dominant factors were the population and social activities. The increasing population has increased demand for housing and venues for various activities, such as residential, industrial, and trading areas. The environmental impact on mangrove forests is an example of land use endangered by various activities.

### 3.2. Mangrove Area Changes in Surabaya

Temperature changes are closely related to changes in land function or use. The land use for mangrove forests in Surabaya is at risk due to population increase. The mangrove forest that functioned as a buffer zone was in critical condition. The condition is illustrated in Table 4.

**Table 4.** Changes in mangrove area in Surabaya in 1996–2016.

| No. | Changes in Mangrove Area per Year (km²) | | | | |
| --- | --- | --- | --- | --- | --- |
|  | **1996** | **2001** | **2006** | **2011** | **2016** |
|  | 1231 | 3304 | 2872 | 4484 | 9459 |

Source: image-processing results, 2016.

The mangrove area in 1996 was 1231 km² and expanded to 3304 km² in 2001. However, the area cover was declined in 2006 to 2872 km² before increasing to 4484 km² in 2011 and 9459 km² in 2016. Changes in mangrove area in Surabaya are shown in Figure 4.

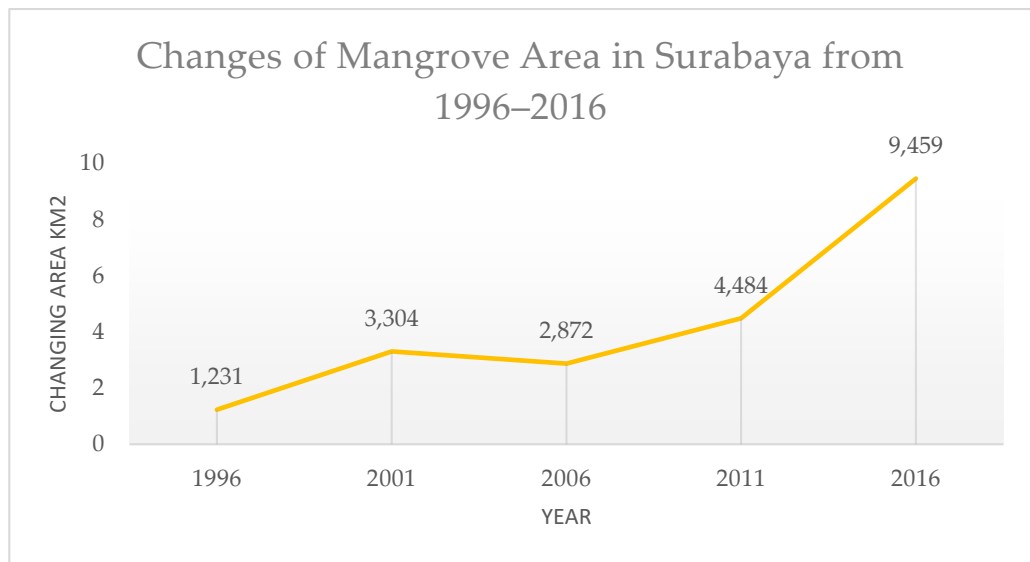

**Figure 4.** Changes in mangrove area in Surabaya in 1996–2016 (km²).

Figure 4 shows that changes in the mangrove area from 1996 to 2016 increased. The highest increase occurred in 2016, when the area was 9459 km².

Currently existing mangrove forests are found in rivers and estuaries or coastal areas. Mangroves grow along rivers and hillsides in the countryside, while mangroves in estuaries and coastal areas are typically fertile, growing along the shoreline. Spatial changes in the mangrove area in Wonorejo and the survey results are shown in Table 5.

Spatial changes in mangrove areas in Surabaya in 1996–2016 were related to the trend of temperature change. Temperature tends to be high in open land or a narrowing mangrove area. Such changes are shown in Table 6.

**Table 5.** Spatial changes in mangrove area.

| No. | Location | Coordinate Point | | Changes in Vegetation Density ($NDVI$ = 0.1–1) | | | | |
|---|---|---|---|---|---|---|---|---|
| | | X | Y | 1996 | 2001 | 2006 | 2011 | 2016 |
| 1. | Parking lot of Mangrove tourism area in Wonorejo | 701,500 | 9,191,892 | 0.516 | 0.509 | 0.594 | 0.368 | 0.481 |
| 2. | Open field (shrubland) in tourism area | 701,924 | 9,191,725 | 0.482 | 0.433 | 0.677 | 0.571 | 0.632 |
| 3. | Pond in tourism area | 702,494 | 919,578 | 0.359 | 0.312 | 0.397 | 0.217 | 0.314 |

**Table 5.** *Cont.*

| No. | Location | Coordinate Point | | Changes in Vegetation Density (*NDVI* = 0.1–1) | | | | |
|-----|----------|---|---|---|---|---|---|---|
| | | X | Y | 1996 | 2001 | 2006 | 2011 | 2016 |
| 4. | Estuary in tourism area | 703,585 | 9,192,102 |  0.161 |  0.195 |  0.125 |  0.162 |  0.111 |
| 5. | Mangrove forest measurement point | 703,614 | 9,192,005 |  0.187 |  0.424 |  0.478 |  0.793 |  0.879 |

Source: Data analysis, 2017.

**Table 6.** Trend of temperature changes in mangrove forest area in Surabaya.

| No | Location | Coordinate Point | | Surface Temperature Change (°C) | | | | |
|---|---|---|---|---|---|---|---|---|
| | | X | Y | 1996 | 2001 | 2006 | 2011 | 2016 |
| 1. | Parking lot of Mangrove tourism area in Wonorejo | 701,500 | 9,191,892 | 24.3 | 29.6 | 27.7 | 27.5 | 30.3 |
| 2. | Open field (shrubland) in tourism area | 701,924 | 9,191,725 | 24.3 | 29.5 | 27.3 | 27.4 | 28.7 |
| 3. | Pond in tourism area | 702,494 | 919,578 | 23.4 | 28.1 | 28.4 | 28.5 | 31.5 |
| 4. | Estuary in tourism area | 703,585 | 9,192,102 | 23.4 | 27.1 | 27.4 | 27.6 | 27.4 |
| 5. | Mangrove forest measurement point | 703,614 | 9,192,005 | 23.8 | 27.4 | 26.1 | 26.6 | 27.9 |

Source: Data analysis, 2017.

### 3.3. Mangrove Forest Ability to Absorb Carbon Dioxide ($CO_2$) in Surabaya Regency

The distribution of $CO_2$ in Surabaya showed that the 40 samples had distinct characteristics depending on land use or covering. The lowest $CO_2$ distribution was 352 ppm, and the highest was 587 ppm. The identification results of $CO_2$ distribution in Surabaya are presented in Table 7.

**Table 7.** $CO_2$ distribution in Surabaya based on survey results.

| No | Classification | $CO_2$ (ppm) | Number of Distribution | Percentage (%) | Land Use |
|---|---|---|---|---|---|
| 1 | Extremely low | 352–394 | 1 | 2.5 | Estuary |
| 2 | Low | 395–446 | 0 | 0 | – |
| 3 | Moderate | 447–493 | 1 | 2.5 | Estuary, mangrove |
| 4 | High | 494–540 | 11 | 27.5 | Pond, industry, cemetery, zoo, golf course, campus, residential area |
| 5 | Extremely high | 541–587 | 27 | 67.5 | Road, industry, residential area, commercial area |
| | Total | | 40 | 100 | |

Source: Data analysis, 2017.

Table 7 shows that $CO_2$ distribution in Surabaya was categorized as high and extremely high at 27.5% and 67.5%, respectively. Only one sample point was found in the moderate and low categories, with a percentage of 2.5%. $CO_2$ distribution in Surabaya was categorized as extremely high. This indicated that Surabaya had already experienced the greenhouse effect, as shown by the connection between temperature data and $CO_2$ levels in the air. The map of $CO_2$ spatial distribution in Surabaya is shown in Figure 5.

The highest $CO_2$ percentage was often found in developed areas with land uses such as residential and industrial areas, roadways, dry ponds, cemeteries, and stations. Figure 5 shows that high temperatures and high $CO_2$ percentages in Surabaya were found in developed areas. Generally, $CO_2$ percentage was low in vegetation cover and waters. Though the $CO_2$ percentage in mangrove areas was categorized as moderate, it was still lower than that in residential or other developed areas. This is supported by a comparison of the $CO_2$ percentage in mangrove forests and developed areas. The $CO_2$ distribution in Surabaya is shown in Table 8.

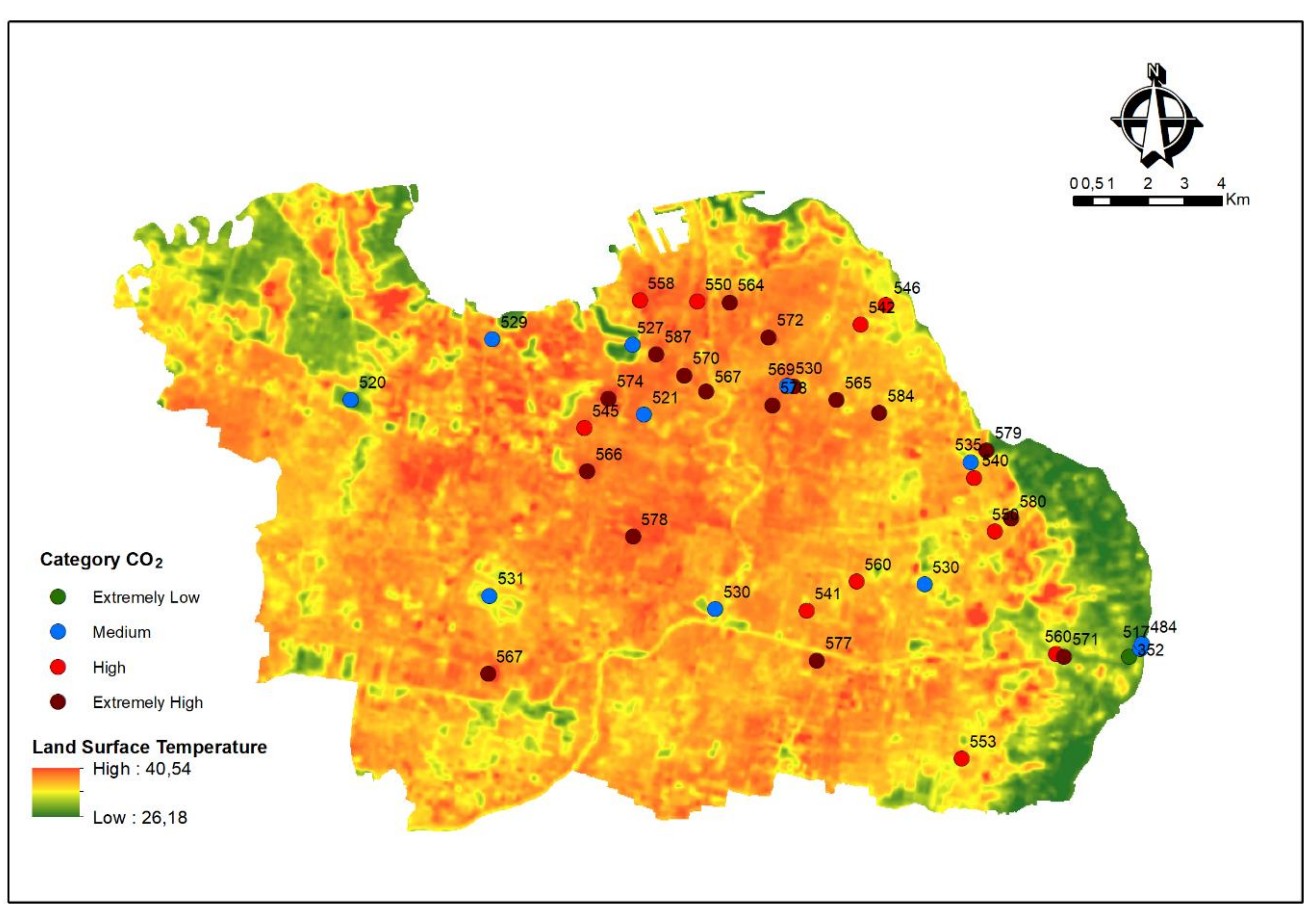

**Figure 5.** Map of $CO_2$ distribution compared with surface temperature in Surabaya at 2016.

**Table 8.** $CO_2$ distribution in Surabaya.

| No | Land Use | $CO_2$ (ppm) | No | Land Use | $CO_2$ (ppm) | No | Land Use | $CO_2$ (ppm) | No | Land Use | $CO_2$ (ppm) |
|---|---|---|---|---|---|---|---|---|---|---|---|
| 1 | Estuary 1 | 352 | 11 | Golf course | 531 | 21 | Container port | 558 | 31 | Fishing pond | 571 |
| 2 | Estuary 2 | 484 | 12 | Residential area | 535 | 22 | Mangrove port of Wonorejo | 560 | 32 | Station | 572 |
| 3 | Mangrove Forest | 517 | 13 | Industrial area | 540 | 23 | Residential area | 560 | 33 | Highway | 574 |
| 4 | Mangrove Forest | 520 | 14 | Soekarno street | 541 | 24 | Business area | 564 | 34 | Residential area | 577 |
| 5 | Pond | 521 | 15 | Industrial area | 542 | 25 | Industrial area | 565 | 35 | Residential area | 578 |
| 6 | Riau street | 527 | 16 | Industrial area | 545 | 26 | Industrial area | 566 | 36 | Residential area | 578 |
| 7 | Industrial area | 529 | 17 | Industrial area | 546 | 27 | Office complex | 567 | 37 | Shrubland | 579 |
| 8 | Cemetery | 530 | 18 | Residential area | 550 | 28 | Industrial area | 567 | 38 | Residential area | 580 |
| 9 | ITS campus | 530 | 19 | Container | 550 | 29 | Residential area | 569 | 39 | Commercial roads Kenjeran housing area | 584 |
| 10 | Zoo | 530 | 20 | Residential area | 553 | 30 | Office complex | 570 | 40 | Office Complex | 587 |

Source: Data analysis, 2017.

## 4. Discussion

The interpretation of the results and the survey of mangrove forest areas showed correlation between land-use changes and increasing surface temperatures. Five sample points in the Wonorejo mangrove forest area showed a significant spatial effect of mangrove conversion into developed areas or open fields. The significant effect was shown by the parameter of observed surface-temperature changes from 1996 to 2016 using Landsat images, and the survey findings were used to validate the temperature. According to a study conducted by Kristoufek [19], there is a significant connection between an increase in $CO_2$ emissions and global temperature.

Changes in the mangrove area had unique stages. This began with the conversion of mangrove areas into ponds. This conversion increased the temperature, as the $CO_2$ absorbed by mangroves increased significantly following the conversion.

Ponds in the mangrove area were explicitly designated as private property. Therefore, individuals can pass or sell ownership rights onto third parties, including property developers.

Mangrove forests have an essential function for Surabaya, serving as a protector against global warming, and a conservation area against erosion and abrasion. Mangrove forests are effective in reducing greenhouse gases, and specifically $CO_2$ [17]. This contributed to the low carbon cycle in the sea latitude, and greenhouse emissions caused by tropical deforestation [9].

The increasing concentration of $CO_2$ has a major effect on plants. Plants can be selected according to if they would survive and adapt to climate change [20]. There should be a preference for mangroves with high $CO_2$ absorption and erosion resistance in deciding on which mangrove types should be created. Mangroves respond differently to geological and temperature conditions [21]. Research found that the carbon percentage in the high-tide zone was higher than that in the low-tide zone [22].

Spatial changes in the mangrove forest area in Surabaya in 1996–2016 were related to temperature changes. Therefore, green areas should be included in spatial planning to control the global-warming effects in Indonesia [23].

Green space-management models should include both green spaces in coastal areas and green spaces that surround commercial routes, railways, industrial sectors, and residential areas [24]. Meanwhile, the use of reflective materials to replant trees in green spaces will increase the greenhouse effect which results in increased global warming and an urban heat island (UHT) effect [25].

Mangrove forests located along the east coast of Surabaya should be maintained as conservation areas. Mangrove forest management is a process for managing global warming [8,26].

## 5. Conclusions

There were changes in the mangrove area and temperature in 1996–2016. The mangrove area expanded in 2016, while temperature increased due to the conversion of mangrove forests into ponds and the development of areas such as airports, residential areas, and factories. $CO_2$ concentrations in waters, mangroves, and developed areas have different characteristics. The highest temperatures were found in residential and industrial areas, and the lowest temperatures were found in estuaries.

This research can be used as a model of micro-temperature management for other developing cities, especially in green spaces management. Furthermore, spatial analysis can estimate changes in mangrove forest to balance the increase in urban areas with the mangrove areas needed.

**Author Contributions:** S.S.: conceptualization, funding acquisition, project administration, writing original draft. P.P.: investigation, resources, software, validation, and visualization. S.B.: conceptualization, data curation, validation, and writing and editing review. All authors have read and agreed to the published version of the manuscript.

**Funding:** Researchers did not use other funds in this research.

**Institutional Review Board Statement:** Not applicable.

**Informed Consent Statement:** Not applicable.

**Data Availability Statement:** Not applicable.

**Acknowledgments:** This study was supported by the Research and Community Services Department of the Ministry of Research, Technology, and Higher Education. We also thank the Social Science Faculty of the State University of Malang, Indonesia.

**Conflicts of Interest:** The authors declare no conflict of interest.

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
