# Peer review of "Spatial Analysis of Mangrove Forest Management to Reduce Air Temperature and CO2 Emissions"

_sustainability, doi:10.3390/su13148090_

Round 1
Reviewer 1 Report
This is an interesting and pertinent topic and some significant findings are presented. However, the flow of the paper is difficult to follow - in part because of the English language used - which needs significant improvement. The charts need to be re-labelled to make it much clearer what they present and what they show.
Finally, the Discussion / Conclusions need considerable work to reflect the data presented and lead to some evidence-based findings. These read as though there is a lot of assertion without reference to the facts of the research findings.
Author Response
Dear reviewer 1
we have revised the article. here we attach the change made

Reviewer 2 Report
The paper "Spatial analysis of mangrove forest management for the reduction of air temperature and CO2 emission" seems to be interesting and relevant because of its topic.
Abstract: the reader from the abstract should already understand what geographic region is represented by "Surabaya". I cannot agree with the statement that "Mangrove forest as a conservation land has an essential function as an oxygen source for people in Surabaya..." - I suppose to look at this process more broadly, not locally; and it is not the only "oxygen source for people".
It is not clear what the authors mean by "... 210-300C had a different trend"
Lacking the degree sign in the "210-300C"
"There was a correlation between mangrove forest change and the temperature change" - do you mean forest cover, species composition?
Overall, all abstract needs much revision.
Introduction:
"earth36/10/2021 7:39:00 AM" - should be corrected ?
line 49: to absorb carbon content - I suggest to use "carbon dioxide"
line 50: "mangroves have a biomass potential of 108.66 and carbon content of 55.35" - units are missing
lines 54-55: what do you mean by "CO"? - should be corrected
line 56: "The carbon assimilation ability of mangrove forest is 4 (four) level higher" - if this is a specific methodology how the level are obtained, then please give the explanation once after this sentence.
line 64: "Plants can emit and absorb unique waves" - you should explain what kind of "waves", give the references, etc.
lines 82-87: this section should be moved to "Materials and Methods" - it is not Introduction....
Better finish the Introduction with some hypothesis, what the authors are expecting from their study, and novelty - are their knowledge gaps that you are investigating this ecosystem, etc.
Materials and Methods:
line 119: the formula should have a number (1), and please explain the indices inside the formula
lines 145; 156, 170 - give the numbers for the formulas - (2) and (3) and (4)
Results:
line 198, Table 3: revise the title of this table - do you measure the temperature by km?? "Annual temperature change (km2)" you should clearly indicate the title; why it is important to give four decimal places?
line 244, Table 4: Changes of Mangrove Area per year (km2) - what do you mean by "changes" - increase? decrease?
always use the "km" not "Km"
line 261, Table 5 - I see "Ecotourism" is mentioned here, I haven't seen it before in the text
line 274, Table 7: "CO2 content distribution" CO2 content should have the units. What is the "Number of Distribution"
Fig 5: it is impossible to distinguish the colors of bubbles of different meanings; you should better use more contrasting colours.
Fig 6: correct CO2; what the blue and red lines mean? I cannon understand how you received the CO2 Content from such land-use like: Riau (what is this??); Cemetery? Jl Soekarno (what is this??)? Office complex? these are not "land-use" (or it is only the very local scale). The authors would better find other entitlement for these "structures". Also, why some of these "land-uses" are mentioned for several times in this table (15 and 25; 27 and 30; etc.)
Conclusions: "The mangrove area became larger in 2016. It was due to the distribution of mangrove to coastal areas because of sedimentation expansion" - but you were not investigating the "sedimentation", it would be better to use other term...
Author Response
Dear reviewer 2
we have revised the article. here we attach the change made

Round 2
Reviewer 1 Report
The paper presents a large amount of what seems to be interesting information and data. However, the main issues remaining are the level of English language and the structure / flow of the presentation and arguments. The paper is very hard to read and comes across as very muddled with lots of typos and grammatical errors. The key points being made / established are not clear and the arguments do not flow consistently through the manuscript.
Similarly the Conclusions are unclear - is it really the case that Mangrove cover has INCREASED due to erosion??
The paper should establish the research question or questions - and clearly state these - at the outset. The methodological approaches employed should then flow from this and lead to the results / analysis / discussion and conclusions etc.
When presenting your abundant findings you need to make clear - what they are and importantly, what they show. You cannot assume that the reader will make sense of this material without effective guidance.
You do need to have a native-speaking English language researcher involved in reviewing this. The structure and logic of the paper need considerable attention.
At the end, tell us what you have found, how that is justified / evidenced / and of this, what is new science.
Reviewer 2 Report
The authors improved the paper according the suggestions.
Author Response
thank you for your feedback.
we have done edit for english error using MDPI proofreader service, also we have followed the suggestions from editor 1